# On the Generalization of Neural Networks Trained with SGD: Information-Theoretical Bounds and Implications

## Abstract

Understanding the generalization behaviour of deep neural networks is an important theme of modern research in machine learning. In this paper, we follow up on a recent work of Neu [49] and present new information-theoretic upper bounds for the generalization error of neural networks trained with SGD. Our bounds and experimental study provide new insights on the SGD training of neural networks. They also point to a new and simple regularization scheme which we show performs comparably to the current state of the art.

## 1   Introduction

The outstanding performance of deep learning has brought to the surface some intriguing properties of deep neural networks, one of which is the observation that despite their high capacity, deep neural networks tend to generalize well [80]. This contradicts classical wisdom in statistical learning theory (e.g., [71] ) and has stimulated intense research interest in understanding the generalization behaviour of modern neural networks.

One theme of research focuses on the study of over-parameterized neural networks, where generalization bounds are obtained [22, 1, 6, 50, 52, 53, 2, 3] and a curious "double descent" phenomenon is observed and analyzed [9, 46, 78]. New bounding techniques for analyzing generalization have also been developed, utilizing information theoretic quantities [64, 65, 77, 5, 13, 69, 23, 7, 4, 30, 84]. The bounds provided by these techniques have the advantages of accounting for both the data structure and the learning algorithm.

The generalization ability of neural networks trained with mini-batched stochastic gradient descent [61], simply referred to SGD in this paper, has also been widely studied. Specifically, built on a connection between stability and generalization [12], a stability-based bound is first presented in [32], followed by a surge of research effort exploiting similar approaches [44, 17, 26, 43, 8]. Information-theoretic bounding techniques have also demonstrated great power in analyzing SGD-like algorithms. For example, [55] is the first to utilize information-theoretical bound in analyzing the generalization ability of SGLD [28, 74]. The bound was subsequently improved by [47, 31, 62, 72]. Inspired by the work of [55], [49] presents an information- theoretic analysis of the models trained with SGD. The analysis of [49] constructs an auxiliary weight process parallel to SGD training and upper-bounds the generalization error through this auxiliary process.

Another line of research connects the generalization of neural networks with the flatness of loss minima [35] found by SGD or its variant [40, 20, 24, 51, 16, 37, 38, 83, 27]. This understanding has led to the discovery of new SGD-based training algorithms for improved generalization. For example, in a concurrent development by [83] and [27], a local "max-pooling" operation is applied to

the loss landscape prior to the SGD updates. This approach, referred to as AMP[83] or SAM[27], is shown to make SGD favor flatter minima and achieve the state-of-the-art performance among various competitive regularization schemes [83].

In this paper, we focus on investigating the generalization of neural networks trained with SGD. We build upon the work of [49]. Following the same construction of the auxiliary weight process in [49], we present upper bounds of generalization error that improve upon [49] in two ways. The first improvement is via removing an unnecessary term in the bounds of [49] by invoking the HWI inequality [60]. The second improvement is via replacing a sample-level mutual information term in [49] with an instance-level mutual information term, exploiting a recent result of [13]. The bounds we obtain decompose into two terms, one measuring the impact of training trajectories ("the trajectory term") and the other measuring the impact of the flatness of the found solution ("the flatness term").

We empirically validate the derived bounds. Various insights are also obtained experimentally concerning the generalization of neural networks under SGD training. For example, the batch size of SGD appears to impact the trajectory term and the flatness term in opposite ways, which complicates the overall dependency of generalization error on batch sizes. A particular interesting observation from our experiments is that a key quantity arising in the trajectory term of the bounds, which we refer to as *gradient dispersion*[1], reveals a double descent phenomenon with respect to training epochs. Most intriguingly, the valley in the double descent curve appears to mark the great divide between the "generalization regime" and the "memorization regime" of training. Furthering from this observation, we also show that it is possible to reduce the memorization effect by dynamically clipping the gradient and reducing its dispersion.

Our bounds also inspire a natural and simple solution to alleviate generalization error. Specifically, we propose a new training scheme, referred to as *Gaussian model perturbation* (GMP), aiming at reducing the flatness term of the bounds. This scheme effectively applies a local "average pooling" to the empirical risk surface prior to SGD, greatly resembling the "max-pooling" approach adopted in AMP[83]. We demonstrate experimentally that GMP achieves a competitive performance with the current art of regularization schemes.

Length constraints precludes elaboration at places. The reader is referred to supplementary materials for proofs and additional information.

**Other Related Literature**  Gradient dispersion is mostly studied from optimization perspectives[11, 63, 39, 75, 25]. Prior to this work, only a few works relate gradient dispersion with the generalization behaviour of the networks. In [49, 72], gradient dispersion also appears in the generalization bounds. In [38], gradient dispersion is argued to capture a notion of "flatness" of the local minima of the loss landscape, thereby correlating with generalization.

Injecting noise in the training process has been proposed in various regularization schemes, for example, [10, 14, 15, 68, 73]. But unlike the Gaussian model perturbation scheme derived in this paper, where noise is injected to the model parameters, noise in those schemes is injected either to the training data or to the network activation.

Gradient clipping is a common technique for preventing gradient exploding (see, e.g., [45, 56]). This technique is also used in [82] to accelerate training. In this paper, gradient clipping is used to investigate and control the impact of gradient dispersion on generalization error.

## 2  Preliminaries

**Population Risk, Empirical Risk and Generalization Error**  Unless otherwise noted, a random variable will be denoted by a capitalized letter (e.g., $Z$), and its realization denoted by the corresponding lower-case letter (e.g. $z$). Let $\mathcal{Z}$ be the instance space of interest and $\mu$ be an unknown distribution on $\mathcal{Z}$, specifying random variable $Z$. Let $\mathcal{W} \subseteq \mathbb{R}^d$ be the space of hypotheses. Suppose that a training sample $S = (Z_1, Z_2, \ldots, Z_n)$ is drawn i.i.d. from $\mu$ and that a stochastic learning algorithm $\mathcal{A}$ takes $S$ as its input and outputs a hypothesis $W \in \mathcal{W}$ according to some conditional distribution $P_{W|S}$ mapping $\mathcal{Z}^n$ to $\mathcal{W}$. Let $\ell : \mathcal{W} \times \mathcal{Z} \to \mathbb{R}^+$ be a loss function, where $\ell(w, z)$ measures the "unfitness" or "error" of any $z \in \mathcal{Z}$ with respect to a hypothesis $w \in \mathcal{W}$. The population risk, for any

---

[1]The quantity is often referred to as gradient variance in the literature [49, 72], but we prefer "dispersion" to "variance" so as to better comply with the mathematical conventions and avoid possible confusion.

$w \in \mathcal{W}$, is defined as

$$L_\mu(w) \triangleq \mathbb{E}_{Z \sim \mu}[\ell(w, Z)].$$

The goal of learning is to find a hypothesis $w$ that minimizes the population risk. But since $\mu$ is only partially accessible via the sample $S$, in practice, we instead turn to the empirical risk, defined as

$$L_S(w) \triangleq \frac{1}{n} \sum_{i=1}^{n} \ell(w, Z_i).$$

The expected generalization error of the learning algorithm $\mathcal{A}$ is then defined as

$$\text{gen}(\mu, P_{W|S}) \triangleq \mathbb{E}_{W,S}[L_\mu(W) - L_S(W)],$$

where the expectation is taken over the joint distribution of $(S, W)$ (i.e., $\mu^n \otimes P_{W|S}$).

Throughout this paper, we take $\ell$ as a continuous function (adopting the usual notion "surrogate loss" [66].). Additionally, we assume that $\ell$ is differentiable almost everywhere with respect to both $w$ and $z$. Furthermore we assume that $\ell(w, Z)$ is $R$-subgaussian[2] for any $w \in \mathcal{W}$. Note that a bounded loss is guaranteed to be subgaussian for all $\mu$ and all $w \in \mathcal{W}$. Let $I(X; Y)$ denote the mutual information [18] between any pair of random variables $(X, Y)$. The following results are known.

**Lemma 1** ([77, Theorem 1.]). *The expected generalization error of algorithm $\mathcal{A}$ is bounded by*

$$|\text{gen}(\mu, P_{W|S})| \leq \sqrt{\frac{2R^2}{n} I(W; S)},$$

**Lemma 2** ([13, Proposition 1.]). *The expected generalization error of algorithm $\mathcal{A}$ is bounded by*

$$|\text{gen}(\mu, P_{W|S})| \leq \frac{1}{n} \sum_{i=1}^{n} \sqrt{2R^2 I(W; Z_i)},$$

**Stochastic Gradient Descent**  We now restrict the learning algorithm $\mathcal{A}$ to be the mini-batched stochastic gradient descent (SGD) algorithm for empirical risk minimization. For each training epoch, the dataset $S$ is randomly split into $m$ disjoint mini-batches, each having size $b$, namely, $n = mb$. Based on each batch, one parameter update is performed. Specifically, let $B_t$ denote the batch used for the $t^{\text{th}}$ update. Define

$$g(w, B_t) \triangleq \frac{1}{b} \sum_{z \in B_t} \nabla_w \ell(w, z),$$

namely, $g(w, B_t)$ is the average gradient computed for the batch $B_t$ with respect to parameter $w$. The rule for the $t^{\text{th}}$ parameter update is then

$$W_t \triangleq W_{t-1} - \lambda_t g(W_{t-1}, B_t),$$

where $\lambda_t$ is the learning rate at the step $t$. The initial parameter setting $W_0$ is assumed to be drawn from the zero-mean spherical Gaussian $\mathcal{N}(0, \sigma_0^2 \mathbf{I}_d)$ with variance $\sigma_0^2$ in each dimension. We will assume that the SGD algorithm stops after $T$ updates and outputs $W_T$ as the learned model parameter.

Given the training sample $S$, let $\xi$ govern the randomness in the sequence $(B_1, B_2, \ldots, B_T)$ of batches. For the simplicity of notion, we will fix the configuration of $\xi$. That is, we will assume a fixed "batching trajectory", or a fixed way to shuffle the example indices $\{1, \ldots, n\}$ and divide them into $m$ batches in each epoch. The presented generalization bounds of this paper can be extended to the case where the batching trajectory is uniformly random (as we set up above). This merely involves averaging over all batching trajectories or taking expectation over $\xi$.

**Auxiliary Weight Process**  We now associate with the SGD algorithm an auxiliary weight process $\{\widetilde{W}_t\}$. Let $\sigma^2$ be given, and let $\sigma_1, \sigma_2, \ldots, \sigma_T$ be a sequence of positive real numbers. Define

$$\widetilde{W}_0 \triangleq W_0, \quad \text{and} \quad \widetilde{W}_t \triangleq \widetilde{W}_{t-1} - \lambda_t g(W_{t-1}, B_t) + N_t, \text{ for } t > 0,$$

where $N_t \sim \mathcal{N}(0, \sigma_t^2 \mathbf{I}_d)$ is a Gaussian noise. The relationship between this auxiliary weight process $\{\widetilde{W}_t\}$ and the weight process $\{W_t\}$ in SGD is shown in the Bayesian network below.

---

[2]Recall that a random variable $X$ is $R$-subgaussian [60] if for any $\rho$, $\log \mathbb{E} \exp\left(\rho\left(X - \mathbb{E}X\right)\right) \leq \rho^2 R^2 / 2$.

$$
\begin{array}{ccccccccc}
& N_1 & & N_2 & & \cdots & & N_{T-1} & & N_T \\
& \downarrow & & \downarrow & & & & \downarrow & & \downarrow \\
\widetilde{W}_0 & \rightarrow & \widetilde{W}_1 & \rightarrow & \widetilde{W}_2 & \rightarrow & \cdots & \rightarrow & \widetilde{W}_{T-1} & \rightarrow & \widetilde{W}_T \\
\| & \nearrow & & \nearrow & & \nearrow & & & & \nearrow & \\
W_0 & \rightarrow & W_1 & \rightarrow & W_2 & \rightarrow & \cdots & \rightarrow & W_{T-1} & \rightarrow & W_T
\end{array}
$$

Let $\Delta_t = \sum_{\tau=1}^{t} N_\tau$. Noting that the weight updates in $\{\widetilde{W}_t\}$ uses the same gradient signal as that used in $\{W_t\}$ (which depends on $W_{t-1}$ not $\widetilde{W}_{t-1}$), it is immediate that $\widetilde{W}_t = W_t + \Delta_t$. Note that this auxiliary process follows the same construction as [49], which we will use to study the generalization error of SGD.

To that end, define *gradient dispersion* at parameter $w$ by

$$
\mathbb{V}(w) \triangleq \mathbb{E}\left[ \|\nabla_w \ell(w, Z) - \mathbb{E}\left[\nabla_w \ell(w, Z)\right]\|_2^2 \right],
$$

where the expectation is taken over $Z \sim \mu$.

For a given sample $s \in \mathcal{Z}^n$, define

$$
\gamma(w, s) \triangleq \mathbb{E}\left[ L_s(w + \Delta_T) - L_s(w) \right],
$$

where the expectation is taken over $\Delta_T$ and $L_s(w)$ is the empirical risk of $s$ at parameter $w$.

In the remainder of the paper, let $S'$ denote another sample drawn from $\mu^n$, independent of all other random variables. The main generalization bound in [49] is re-stated below.

**Lemma 3** ([49, Theorem 1.]). *The generalization error of SGD is upper bounded by*

$$
|\mathrm{gen}(\mu, P_{W_T|S})| \leq \sqrt{\frac{2R^2}{n} \sum_{t=1}^{T} \frac{\lambda_t^2}{\sigma_t^2} \mathbb{E}\left[ \Psi(W_{t-1}) + \frac{\mathbb{V}(W_{t-1})}{b} \right]} + |\mathbb{E}\left[\gamma(W_T, S) - \gamma(W_T, S')\right]|,
$$

*where* $\Psi(w_{t-1}) \triangleq \mathbb{E}\left[\|\nabla_w \ell(w_{t-1}, Z) - \nabla_w \ell(w_{t-1} + \zeta, Z)\|_2^2\right]$ *and* $\zeta \sim \mathcal{N}(0, 2\sum_{i=1}^{t-1} \sigma_i^2 \mathrm{I}_d)$.

The term $\Psi(w_{t-1})$ in the bound is referred to as "local gradient sensitivity" in [49].

## 3 New Generalization Bounds for SGD

We first prove that the generalization bound in Lemma 3 can be tightened by removing the local gradient sensitivity term $\Psi(w_{t-1})$. The key observation is that an independence condition used for establishing Lemma 3 in [49] is unnecessary (see Lemma 4 in [49]). This requires invoking a vector version of the HWI inequality [60, Lemma 3.4.2], which we prove in this paper.

**Lemma 4.** *Let X and Y be two random vectors in $\mathbb{R}^d$, and let $N \sim \mathcal{N}(0, \mathbf{I}_d)$ be independent of $(X, Y)$. Then, for every $t > 0$,* $\mathrm{D}_{\mathrm{KL}}(P_{X+\sqrt{t}N} \| P_{Y+\sqrt{t}N}) \leq \frac{1}{2t} \mathbb{E}\left[\|X - Y\|^2\right]$.

Here $\mathrm{D}_{\mathrm{KL}}$ is the KL divergence. Note that the bound in Lemma 3 relies on a similar result which however requires the independence of $X$ and $Y$. Using Lemma 4, we obtain the following theorem.

**Theorem 1.** *The generalization error of SGD is upper bounded by*

$$
|\mathrm{gen}(\mu, P_{W_T|S})| \leq \sqrt{\frac{2R^2}{nb} \sum_{t=1}^{T} \frac{\lambda_t^2}{\sigma_t^2} \mathbb{E}\left[\mathbb{V}(W_{t-1})\right]} + |\mathbb{E}\left[\gamma(W_T, S) - \gamma(W_T, S')\right]|.
$$

The proof of this theorem, as the bounds in [49], relies on Lemma 1 and the sample-level mutual information bound therein. This theorem can be further tightened by exploiting the fact that the instance-level mutual information bound in Lemma 2 is in fact tighter than the sample-level mutual information bound in Lemma 1, as shown in [13]. The main ingredient to proceed in this direction is the following lemma.

**Lemma 5.** *Let $G_t = -\lambda_t g(W_{t-1}, B_t)$. If $Z_i \in B_t$, then $I(G_t + N_t; Z_i | \widetilde{W}_{t-1}) \leq \frac{\lambda_t^2}{\sigma_t^2 b^2} \mathbb{E}\left[\mathbb{V}(W_{t-1})\right]$.*

In this lemma, the mutual information $I(G_t + N_t; Z_i|\widetilde{W}_{t-1})$ roughly indicates the degree by which the SGD's updating signal $G_t$ (smoothed with noise) depends on an individual training instance $Z_i$, when $Z_i$ is used for computing the gradient. When this dependency is strong (giving rise to a high value of the mutual information), the model conceivably tends to overfit the individual training instances. This lemma suggests that the strength of this dependency can be upper-bounded by the expected gradient dispersion at the current weight configuration. In our experiments, we will estimate the expected gradient dispersion and validate this intuition.

It is remarkable that the noise $\{N_t\}$ plays an important role for the bound to hold. To see this, consider $b = 1$ and $\mathcal{Z}$ is countable and large. Then $I(G_t; Z_t|W_{t-1})$ is merely the conditional entropy $H(Z_i|W_{t-1})$, which would grow with sample size $n$ at least as $\log n$. Upper-bounding it with a quantity independent of $n$ would be impossible – This justifies the construction of the auxiliary weight process.

We now state our main theorem. Unlike Theorem 1, which considers a random batching trajectory, this theorem considers a fixed batching trajectory to keep the expression less cluttered. For that batching trajectory, we will use $\mathcal{T}_i$ to denote the set of indices of batches $B_t$ containing instance $Z_i$.

**Theorem 2.** *The expected generalization error of SGD is bounded by*

$$|\mathrm{gen}(\mu, P_{W_T|S})| \leq \frac{R}{nb} \sum_{i=1}^{n} \sqrt{\sum_{t \in \mathcal{T}_i} \frac{2\lambda_t^2}{\sigma_t^2} \mathbb{E}\left[\mathbb{V}(W_{t-1})\right]} + |\mathbb{E}\left[\gamma(W_T, S) - \gamma(W_T, S')\right]|.$$

With an additional assumption, the second term in the bound can be re-expressed, as shown in the following corollary.

**Corollary 1.** *Assume $L_\mu(w_T) \leq \mathbb{E}_\Delta\left[L_\mu(w_T + \Delta_T)\right]$, then the following holds,*

$$\mathrm{gen}(\mu, P_{W_T|S}) \leq \frac{R}{nb} \sum_{i=1}^{n} \sqrt{\sum_{t \in \mathcal{T}_i} \frac{2\lambda_t^2}{\sigma_t^2} \mathbb{E}\left[\mathbb{V}(W_{t-1})\right] + \frac{\sum_{t=1}^{T} \sigma_t^2}{2} \mathrm{Tr}\left(\mathbb{E}\left[\mathrm{H}_{W_T}(Z)\right]\right)},$$

*where $\mathrm{H}_{W_T}$ is the Hessian matrix of the loss with respect to $W_T$ and $\mathrm{Tr}(\cdot)$ denotes trace.*

Corollary 1 follows directly from the second order Taylor expansion of the second term in the bound of Theorem 2. The condition $L_\mu(w_T) \leq \mathbb{E}_\Delta\left[L_\mu(w_T + \Delta_T)\right]$ indicates that the perturbation does not decrease the population risk. This is also assumed in [27] in the derivation of a PAC-Bayesian generalization bound.

Notably, in the bound of Theorem 2, the first term captures the impact of the training trajectory ("trajectory term"), and the second term captures the impact of the final solution. As seen in Corollary 1, this term in fact measures the flatness for the loss landscape at the found solution ("flatness term"). The previous bound of [49] (Lemma 3) and its tightened version in Theorem 1 also similarly contain a trajectory term and a flatness term. Despite that the flatness term there are identical to that in Theorem 2, we now show the trajectory term in Theorem 2 does improve on its counter-part in Theorem 1.

**Lemma 6.** *Assume the instances are sampled without replacement in every epoch. Then the trajectory term in Theorem 2 is upper-bounded by*

$$\min\left\{\frac{R}{n} \sum_{t=1}^{T} \sqrt{\frac{2\lambda_t^2}{\sigma_t^2} \mathbb{E}\left[\mathbb{V}(W_{t-1})\right]}, \sqrt{\frac{2R^2}{nb} \sum_{t=1}^{T} \frac{\lambda_t^2}{\sigma_t^2} \mathbb{E}\left[\mathbb{V}(W_{t-1})\right]}\right\}.$$

The condition in Lemma 6 is usually satisfied in practice. This lemma then immediately implies that the trajectory term in Theorem 2 is no worse than that in Theorem 1. Incorporating this result, if we restrict the smoothness of the loss function $\ell$, we may obtain another version of the generalization bound (although the flatness term therein is expected to be looser than that in Corollary 1).

**Corollary 2.** *If the loss function is differentiable and $\beta$-smooth with respect to $w$, then under the condition of Lemma 6,*

$$|\mathrm{gen}(\mu, P_{W_T|S})| \leq \min\left\{\frac{R}{n} \sum_{t=1}^{T} \sqrt{\frac{2\lambda_t^2}{\sigma_t^2} \mathbb{E}\left[\mathbb{V}(W_{t-1})\right]}, \sqrt{\frac{2R^2}{nb} \sum_{t=1}^{T} \frac{\lambda_t^2}{\sigma_t^2} \mathbb{E}\left[\mathbb{V}(W_{t-1})\right]}\right\} + \beta d \sum_{t=1}^{T} \sigma_t^2.$$

To conclude, we remark that these bounds suggest that in order for the model to generalize well, both the trajectory term and the flatness term need to be small — the former involves the interaction of the learning rate and batch size with the gradient dispersion along the training trajectory, whereas the latter depends on the flatness of the empirical risk surface at the found solution.

# 4 Experimental Study

**Bound Verification** We first verify our bound in Corollary 1 by training an MLP (with one hidden layer) and an AlexNet [42] on MNIST and CIFAR10 [41], respectively. To simplify estimation, we fix the weight initialization and set $\sigma_t$ and $\lambda_t$ to be constants $\sigma$ and $\lambda$, respectively. To compute $\sum_{i=1}^{n} \sqrt{\sum_{t \in \mathcal{T}_i} \mathbb{E}\left[\mathbb{V}(W_{t-1})\right]}$, we compute the gradient dispersion as its empirical estimate from a batch, utilizing a PyTorch [54] library BackPack [19]. To compute $\text{Tr}\left(\mathbb{E}\left[H_{W_T}(Z)\right]\right)$, we randomly sample 10% of the training data and use the PyHessian library [79] to compute the Hessian. Since every choice of $\sigma$ gives a valid generalization bound in Corollary 1, we need to find the optimal $\sigma$, which gives the tightest bound. This can be done by simply utilizing the fact $A/\sigma + \sigma^2 B \geq 3(A/2)^{2/3}B^{1/3}$ for any positive $A$ and $B$, where the equality is achieved by the optimal $\sigma$. We set the sub-gaussian parameter $R = 0.1$. The implementation in this paper is on PyTorch, and all the experiments are carried out on NVIDIA Tesla V100 GPUs (32 GB).

We perform experiments with varying network width and varying levels of label noise. Specifically, label noise level $\epsilon$ refers to the setting where we replace the labels of $\epsilon$ fraction of the training and testing instances with random labels. The estimated bound is compared against the true generalization gap, namely, the difference between the training loss and testing loss, and is shown in Figure 1.

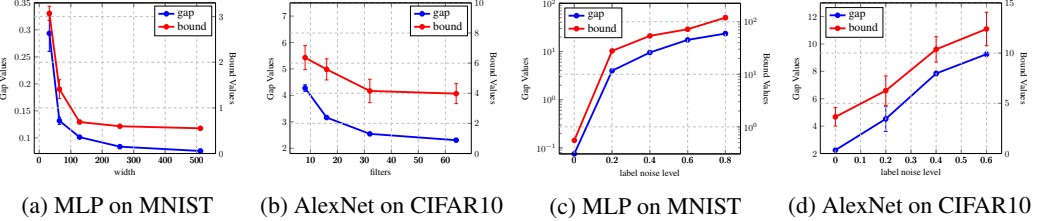

| (a) MLP on MNIST | (b) AlexNet on CIFAR10 | (c) MLP on MNIST | (d) AlexNet on CIFAR10 |

Figure 1: Estimated bound and empirical generalization gap ("gap") as functions of network width ((a) and (b)) and label noise level ((c) and (d)). Left Y-axis: gap value; right Y-axis: bound value.

In Figure 1, we see that in all cases the estimated bound follows closely the trend of the true generalization gap. The fact that the bound curve consistently tracks the gap curve under various label noise levels indicates that our bound very well captures the changes of the data distribution. Note that in Figure 1 (a) and (b), our bound decays with the increase of the model size, showing a trend as opposite to the bounds obtained in classical learning theory. But such a trend clearly better explains the generalization behaviour of modern neural networks.

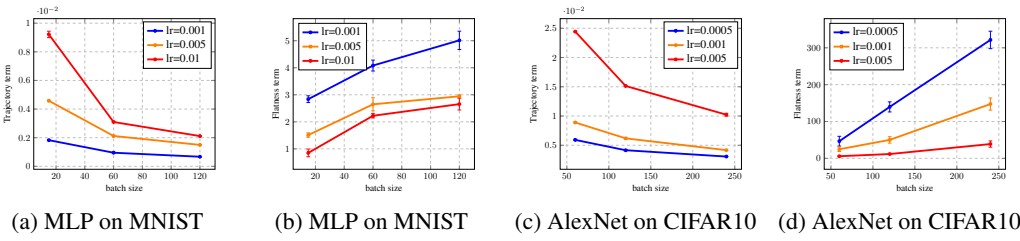

| (a) MLP on MNIST | (b) MLP on MNIST | (c) AlexNet on CIFAR10 | (d) AlexNet on CIFAR10 |

Figure 2: The impact of learning rate and batch size on the trajectory term and the flatness term.

**Learning Rate and Batch Size.** The learning rate and batch size in SGD have explicitly appeared in the trajectory term of the bound in Theorem 2. From the way they appear in the bound, one may be tempted to assert that a small learning rate or large batch size will improve generalization. This would then contradict some previous observations [37, 76, 33], in which increasing the ratio of learning rate

to batch size will benefit generalization. We now investigate this by performing experiments with varying learning rates and batch sizes. In our experiments, the model is continuously updated until the average training loss drops below 0.0001. We separate trajectory and flatness terms of the bound and plot them in Figure 2.

A key observation in Figure 2 is that the learning rate impacts the trajectory term and the flatness term in opposite ways, as seen, for example, in (a) and (b), where the two set of curves swap their orders in the two figures. On the other hand, the batch size also impacts the two terms in opposite ways, as seen in (a) and (b) where curves decrease in (a) but increase in (b). This makes the generalization bound, i.e., the sum of the two terms, have a rather complex relationship with the settings of learning rate and batch size. This relationship is further complicated by the fact that a small learning rate requires a longer training time, or a larger number $T$ of training iterations, which increases the number that are summed over in the trajectory term. Nonetheless, we do observe that a smaller batch size gives a lower value of the flatness term ((b) and (d)), confirming the previous wisdom that small batch sizes enable the neural network to find a flat minima [40].

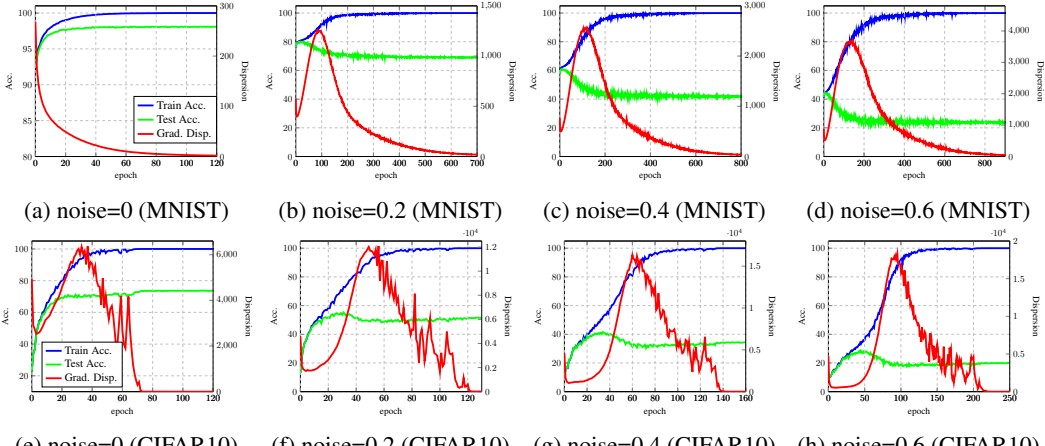

|  |  |  |  |
|---|---|---|---|
| (a) noise=0 (MNIST) | (b) noise=0.2 (MNIST) | (c) noise=0.4 (MNIST) | (d) noise=0.6 (MNIST) |
| (e) noise=0 (CIFAR10) | (f) noise=0.2 (CIFAR10) | (g) noise=0.4 (CIFAR10) | (h) noise=0.6 (CIFAR10) |

Figure 3: Epoch-wise double descent of gradient dispersion, in relation to training/testing accuracies.

**Double Descent of Gradient Dispersion** We experimentally investigate the impact of gradient dispersion on the training of the neural networks by fixing the learning rate, batch size and weight initialization for the each model (MLP for MNIST, AlexNet for CIFAR10). For each model and various label noise levels, we plot in Figure 3 the evolution of the (empirical) gradient dispersion $\widehat{\mathbb{V}}(w_t)$, training accuracy and testing accuracy across training epochs.

An intriguing epoch-wise "double descent" phenomenon is observed, particularly when the labels are noisy. According to the double descent curve, the training may be split into three phases (e.g., Figure 3 (h)). In the first phase, the gradient dispersion rapidly descends and maintains a very low level. In this phase, both training and test accuracies increase while maintaining a very small generalization gap. This suggests that the network in this phase is extracting useful patterns and generalizes well. In the second phase, the gradient dispersion starts increasing until it reaches a peak value. In this phase, the training and testing accuracies gradually diverge, marking the model entering an overfitting or "memorization" regime – when the data contains the noisy labels, the network mostly tries to memorize the labels in the training set. In the third phase, the gradient dispersion descends again, reaching a low value. In this phase, the model continuously overfits the training data, until the training and testing curves reach their respective maximum and minimum. It appears that the timing of the three phases depends on the dataset and the label noise level. For simpler data (e.g. MNIST) and cleaner datasets (e.g. CIFAR10 with low label noise), the first phase may be shorter. This is arguably because in these datasets, extracting useful patterns is relatively easier. Nonetheless, the valley in the double-descent curve appears to mark a "great divide" between generalization and memorization.

**Dynamic Gradient Clipping** Inspired by our generalization bounds and above observations, one way to reduce the generalization error is to control the trajectory term of the bounds by reducing the gradient dispersion in each training step. Here we investigate a simple scheme that dynamically clips

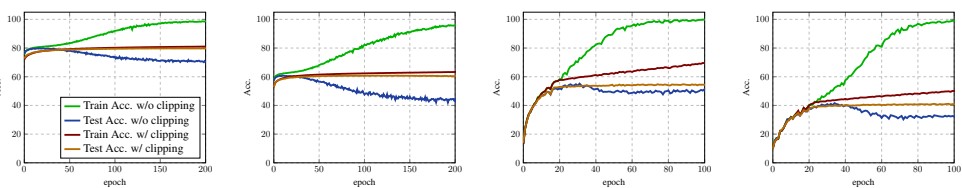

(a) noise=0.2 (MNIST) (b) noise=0.4 (MNIST) (c) noise=0.2 (CIFAR10) (d) noise=0.4 (CIFAR10)

Figure 4: Dynamic Gradient Clipping.

the gradient norm so as to reduce the gradient dispersion. Specifically, whenever the current gradient norm is larger than the gradient norm $K$ steps earlier, or $||g(W_t, B_t)||_2 > ||g(W_{t-K}, B_{t-K})||_2$ (i.e., the model is expected to have entered the "memorization" regime), we reduce the norm of the current gradient $g(W_t, B_t)$ to $\alpha$ fraction of $||g(W_{t-K}, B_{t-K})||_2$, for some prescribed value $\alpha < 1$. The effectiveness of this scheme is best demonstrated when the labels contain noise. As shown in Figure 4, dynamic gradient clipping significantly closes the gap between the training accuracy and the testing accuracy. The models trained with this scheme maintain a near-optimal testing accuracy (e.g., about 80% when the label noise level of MNIST is 0.2), without suffering from the severe memorization effect as seen in models trained without this scheme. Further understanding of the double-descent phenomenon of the gradient dispersion may enable more delicate design of such a dynamic clipping scheme and potentially lead to novel and powerful regularization techniques.

## 5 A Practical Implication: Gaussian Model Perturbation

The appearance of the flatness term in our generalization bounds suggests that for an empirical risk minimizer $w^*$ to generalize well, it is necessary that the empirical risk surface at $w^*$ is flat, or insensitive to a small perturbation of $w^*$. This naturally motivates a training scheme using the following regularized loss:

$$\min_w L_s(w) + \rho \mathop{\mathbb{E}}_{\Delta \sim \mathcal{N}(0, \sigma^2 \mathbf{I}_d)} [L_s(w + \Delta) - L_s(w)],$$

where $\rho$ is a hyper-parameter. Replacing the expectation above with its stochastic approximation using $k$ realizations of $\Delta$ gives rise to the following optimization problem.

$$\min_w \frac{1}{b} \sum_{z \in B} \left( (1 - \rho)\ell(w, z) + \rho \frac{1}{k} \sum_{i=1}^{k} \left( \ell(w + \delta_i, z) \right) \right).$$

We refer to the SGD training scheme using this loss as *Gaussian model perturbation* or GMP. Notably, GMP requires $k + 1$ forward passes for every parameter update. Empirical evidence shows that a small $k$, for example, $k = 3$, already gives competitive performance. Implementing the $k + 1$ forward passes on parallel processors further reduces the computation load.

We experimentally compare GMP with several major regularization schemes in the current art, including Dropout [68], label smoothing [70], Flooding [36], MixUp [81], adversarial training [29], and AMP [83]. The compared schemes are evaluated on three popular benchmark image classification datasets SVHN [48], CIFAR-10 and CIFAR-100 [41]. Two representative deep architectures PreActResNet18 [34] and VGG16 [67] are taken as the underlying model. We train the models for 200 epochs by SGD. The learning rate is initialized as 0.1 and divided by 10 after 100 and 150 epochs. For all compared models, the batch size is set to 50 and weight decay is set to $10^{-4}$. For GMP, we choose $\rho = 0.5$ and set the standard deviation of the Gaussian noise $\Delta$ to 0.03. The value of $k$ is chosen as 3 and 10 respectively (referred to as GMP$^3$ and GMP$^{10}$).

The performances of all compared schemes are given in Table 1. For the compared regularization schemes except GMP, we directly report their performances as given in [83]. Performances of vanilla ERM without regularization are also included as a reference.

Table 1 demonstrates the effectiveness of GMP. Overall GMP performs comparably to the current art of regularization schemes, although appearing slightly inferior to the most recent record given by AMP [83]. Noting that the key ingredient of AMP, "max-pooling" in the parameter space, greatly resembles regularization term in GMP, which may be seen as "average-pooling" in the same space.

| PreActResNet18 | Top-1 Acc. (%) | PreActResNet18 | Top-1 Acc. (%) | PreActResNet18 | Top-1 Acc. (%) |
|---|---|---|---|---|---|
| ERM | 97.05±0.063 | ERM | 94.98±0.212 | ERM | 75.69±0.303 |
| Dropout | 97.20±0.065 | Dropout | 95.14±0.148 | Dropout | 75.52±0.351 |
| Label Smoothing | 97.22±0.087 | Label Smoothing | 95.15±0.115 | Label Smoothing | 77.93±0.256 |
| Flooding | 97.16±0.047 | Flooding | 95.03±0.082 | Flooding | 75.50±0.234 |
| MixUp | 97.26±0.044 | MixUp | **95.91±0.117** | MixUp | **78.22±0.210** |
| Adv. Training | 97.23±0.080 | Adv. Training | 95.01±0.085 | Adv. Training | 74.77±0.229 |
| AMP | **97.70±0.025** | AMP | **96.03±0.091** | AMP | **78.49±0.308** |
| **GMP**[3] | **97.43±0.037** | **GMP**[3] | 95.64±0.053 | **GMP**[3] | 78.05±0.208 |
| **GMP**[10] | 97.34±0.058 | **GMP**[10] | 95.71±0.073 | **GMP**[10] | 78.07±0.170 |
| VGG16 | Top-1 Acc. (%) | VGG16 | Top-1 Acc. (%) | VGG16 | Top-1 Acc. (%) |
| ERM | 96.86±0.060 | ERM | 93.68±0.193 | ERM | 72.16±0.297 |
| Dropout | 97.04±0.049 | Dropout | 93.78±0.147 | Dropout | 72.28±0.337 |
| Label Smoothing | 96.93±0.070 | Label Smoothing | 93.71±0.158 | Label Smoothing | 72.51±0.179 |
| Flooding | 96.85±0.085 | Flooding | 93.74±0.145 | Flooding | 72.07±0.271 |
| MixUp | 96.91±0.057 | MixUp | **94.52±0.112** | MixUp | 73.19±0.254 |
| Adv. Training | 97.06±0.091 | Adv. Training | 93.51±0.130 | Adv. Training | 70.88±0.145 |
| AMP | **97.27±0.015** | AMP | 94.35±0.147 | AMP | 74.40±0.168 |
| **GMP**[3] | **97.18±0.057** | **GMP**[3] | 94.33±0.094 | **GMP**[3] | **74.45±0.256** |
| **GMP**[10] | 97.09±0.068 | **GMP**[10] | **94.45±0.158** | **GMP**[10] | **75.09±0.285** |
| (a) SVHN | | (b) CIFAR-10 | | (c) CIFAR-100 | |

Table 1: Top-1 classification accuracy on (a) SVHN, (b) CIFAR-10 and (c) CIFAR-100. We run experiments 10 times and report the mean and the standard deviation of the testing accuracy.

## 6 Conclusion and Outlook

This paper presents new generalization bounds for neural networks trained with SGD, improving upon the results of [49]. Our bounds naturally point to new and effective regularization schemes. At the same time, they reveal interesting phenomena in the SGD training of neural networks. While these phenomena deserve further investigation in their own right, we here suggest another direction for improving the bounds, namely, via the use of strong data-processing inequalities (DPI) [57–59] (noting that the standard DPI is in fact needed for establishing Theorem 2).

For any Markov chain $U \to X \to Y$, we will denote by $\mathcal{U}$, $\mathcal{X}$, and $\mathcal{Y}$ the spaces in which $U$, $X$, $Y$ take values, respectively. For any distribution $P$ on $\mathcal{X}$, we will use $P_{Y|X} \circ P$ to denote the distribution on $\mathcal{Y}$ induced by the push-forward of the distribution $P$ by $P_{Y|X}$, namely, for any $y \in \mathcal{Y}$, $\left(P_{Y|X} \circ P\right)(y) \triangleq \int P_{Y|X}(y|x)P(x)dx$. Let $\mathcal{S}(U)$ be the support of $P_U$ and $\mathcal{H}(U, P_{X|U})$ be the convex hull of $\{P_{X|U=u} : u \in \mathcal{S}(U)\}$. Define

$$\eta(U \to X \to Y) \triangleq \sup_{P,Q \in \mathcal{H}(U, P_{X|U})} \frac{\mathrm{D}_{\mathrm{KL}}(P_{Y|X} \circ P || P_{Y|X} \circ Q)}{\mathrm{D}_{\mathrm{KL}}(P||Q)}$$

**Lemma 7.** *For any Markov chain $U \to X \to Y$, $I(U;Y) \leq \eta(U \to X \to Y)I(U;X)$.*

Here $\eta(U \to X \to Y)$ serves as the "contraction coefficient" for the stochastic kernel $P_{Y|X}$, characterizing the greatest extent by which the kernel may bring closer any two distributions on $\mathcal{S}(U)$ in its output space. It is easy to see that $\eta(U \to X \to Y) \leq 1$, giving rise to a stronger DPI.

Denote $V_t \triangleq \widetilde{W}_{t-1} + G_t$. It can be verified that $Z_i \to V_t \to \widetilde{W}_t$ form a Markov chain. Denote $\eta_{i,t} \triangleq \eta(Z_i \to V_t \to \widetilde{W}_t)$, and $\Gamma_i^t = \{t+1, t+2, \ldots, T\} \setminus \mathcal{T}_i$. Theorem 2 can be improved to:

**Theorem 3.** *The expected generalization error of SGD is bounded by*

$$|\mathrm{gen}(\mu, P_{W_T|S})| \leq \frac{2R}{nb} \sum_{i=1}^{n} \sqrt{\sum_{t \in \mathcal{T}_i} \frac{\lambda_t^2}{\sigma_t^2} \mathbb{E}\left[\mathbb{V}(W_{t-1})\right] \cdot \prod_{\tau \in \Gamma_i^t} \eta_{i,\tau}} + |\mathbb{E}\left[\gamma(W_T, S) - \gamma(W_T, S')\right]|.$$

It remains to characterize the contraction coefficient $\eta_{i,\tau}$ in a computable form. Simply bounding it via the Dobrushin's coefficient [21], as suggested in [72] for analyzing SGLD, is unlikely to make the bound in this theorem significantly tighter than that in Theorem 2.

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
