# OpenReview forum: "On the Generalization of Neural Networks Trained with SGD: Information-Theoretical Bounds and Implications"
_NeurIPS.cc/2021/Conference — NeurIPS 2021 Submitted_

### Official Review · Reviewer_dXbz · 2021-07-15

**Rating:** 4
**Confidence:** 4

**Summary:**


This paper considers the important problem of understanding the generalization of the SGD algorithm. This paper is a follow up on this paper:

 G. Neu. “Information-theoretic generalization bounds for stochastic gradient descent”.

To analyze the generalization, the authors use the mutual information framework of Russo-Zou and Xu-Raginsky where the expected generalization error is upper bounded using mutual information (MI) between output and training set. The main issue with analyzing SGD with MI is that it leads to vacuous bounds due to the fact that there is only degenerate noise in SGD. To circumvent this issue,  the authors use the idea of analyzing a “surrogate algorithm” from Neu’ 21 paper. The surrogate algorithm is a noisy version of the SGD, and instead of analyzing the SGD we can analyze the generalization of the surrogate algorithm. At the end we need to add “pay” for the difference between the output of the surrogate algorithm and SGD.


**Ethical Concerns:**

There is no specific ethical concerns with this paper.

**Limitations And Societal Impact:**

I think the limitations of this work is not discussed thoroughly in the paper. The authors can refer to my questions above to find suggestions for improvement.

**Main Review:**

Contribution and Positive points:

The main contributions of the authors on the top of Neu’ 21 paper are obtaining a tighter bound using the mutual information bound of Bu et al. in Theorem 2. Note that the construction of the surrogate algorithm is exactly the same as the paper by Neu'21.
An interesting point about the main results is that the generalization bound both depends on the trajectory of SGD and the final solution. In particular, the variance of the gradients along the pass captures how trajectory affects the generalization. Then, in Corollary 2 they have a nice result which shows the sharpness of the final solution impacts the generalization. Finally, they have an extensive empirical study of their bound on different benchmarks.

However, I have some major comments and some of them are concerned with the validity of their proof.

Weakness and Questions:

-I think the introduction needs to be revised. It is not clear how this work relates to other work in the literature of understanding generalization of SGD. Note the problem of understanding the generalization of SGD has been approached using lots of different tools and techniques.

-General comment about the paper: In the abstract and in the introduction, it has been emphasized that this paper aims to derive generalization bounds for neural networks trained with SGD. However, it is not clear how the quantities appearing in the main theorem in the paper (Theorem 2) depend on the neural networks parameters. Specifically, I could not find anything specific about neural networks in the main results. In fact the results hold for any learning objective which is optimized with SGD. This is in contrast with some of the known results on the generalization of neural networks which connect the generalization to the specific parameters of the neural networks.

-Corollary 2: One important issue with this result is the dependence on the dimension. This is in contrast with the classical analysis of SGD using the stability framework which is dimension-free. How do you compare the stability method and your method?

-Corollary 1: Assume we want to study the generalization error of a NN trained with SGD with respect to 0-1 loss. 0-1 loss is subG with parameter ¼. However, it is not continuous. Therefore, the bound in corollary 2 can’t be used for 0-1 loss function since it does not have second-derivative. Because the Gamma term is defined with respect to the loss function. How does this affect the conclusion of the paper on the impact of the flatness of the final solution on the generalization?

-Diagonal covariance: Adding gaussian noise seems to be reasonable. However, why do not you consider the non-diagonal covariance?

-Stochastic Convex optimization: Assume that the loss function is convex, lipschitz, and smooth. How can you use this information to design the noise variances? It seems to me that this framework in this regime is much looser than Hardt et al results specifically because of the dependence on the dimension.

-Line 205: In your numerical results what do you mean by setting the sub-gaussian parameter to 0.1? If you use 0-1 loss then the subgaissian parameter is ¼.

-It seems you use corollary 1 for empirical studies. As pointed out, this is not valid, it is not valid for 0-1 loss function.

Comments of the proof:


-The conditional expectation notation in the paper should be revised. For instance in Eq (9) the notation is not correct and it makes it very difficult for readers to understand it. The outer expectation is over S,S’ but inside there is a conditioning on the specific realizations. The same error appears in the rest of the proof.

-I have difficulty understanding  Eq(6) - Eq(10). Specifically, what do you mean by “we do not use an independent copy of \tilde{W}”?  I believe the derivation in Neu’ 21 is correct (beginning of Page 8 in the V1 arxiv version of Neu’ 21). My understanding is as follows: P(W|S) is a kernel which shows the learning algorithm: specifically given a training set, it outputs a distribution on W. Therefore, when you have two independent training sets S and S’, they will generate two independent copies of W and W’ whose law are given by P(W|S) and P(W’|S’). Specifically Eq(8) in your paper is not correct. This is an extremely serious issue and I would like to hear your detailed response on this.

-Could you please elaborate on Eq (11)? Eq(10) follows from the monotonicity of the KL divergence or Data Processing. In Eq (11) you wanted to use the chain rule for KL. But the caveat here is that the surrogate algorithm \tilde{W} is NOT Markovian as shown in the figure on Page 4 of your paper.  As a result,  Eq (11) in its current form is NOT CORRECT. I would appreciate it if you could comment on it since it is an important step in the proof and it might affect the validity of the proof.

-I think in Eq. 11 you wanted to use the tower rule but you forgot to write the outer expectation.


**Time Spent Reviewing:**

3

---

> ### Author Response · Authors · 2021-08-10
> **To Reviewer dXbz**
>
> Thank you very much for your careful reading and constructive comments. Our responses follow.
>
> - I think the introduction needs to be revised. It is not clear how this work relates to other work in the literature of understanding generalization of SGD. Note the problem of understanding the generalization of SGD has been approached using lots of different tools and techniques.
>
> **Response.**	We will make our best effort in the revision to better position this work in the landscape of other related literature.
>
> - General comment about the paper: In the abstract and in the introduction, it has been emphasized that this paper aims to derive generalization bounds for neural networks trained with SGD. However, it is not clear how the quantities appearing in the main theorem in the paper (Theorem 2) depend on the neural networks parameters. Specifically, I could not find anything specific about neural networks in the main results. In fact the results hold for any learning objective which is optimized with SGD. This is in contrast with some of the known results on the generalization of neural networks which connect the generalization to the specific parameters of the neural networks.
>
> **Response.**	Yes, our theorems are applicable to any learning objective optimized by SGD. But the empirical verification of this paper focuses on the generalization of DNN trained by SGD. Agreeably developing neural network specific generalization bounds is of great importance. But presenting more general results that are applicable beyond neural networks should not be taken negatively. We will modify the abstract and introduction to better scope this work so that they are consistent with the rest of the paper.
>
>
>
> - Corollary 2: One important issue with this result is the dependence on the dimension. This is in contrast with the classical analysis of SGD using the stability framework which is dimension-free. How do you compare the stability method and your method?
>
> **Response.** The dependence of dimension only arises in the bound in Corollary 2, where the flatness term is upper-bounded through a smoothness parameter and the model dimension $d$. Note that in the main theorem (Theorem 2) and Corollary 1, the flatness term does not explicitly depends on $d$. For example, the trace of the Hessian in Corollary 1 captures the local geometry of the loss landscape at the found minima, which needs not to depend on $d$. This can be seen in Figure 1, in which as the model increases its width, the generalization bound in Corollary 1 decreases.
>
> Coming back to Corollary 2, we note that the dependence of $d$ in the bound results from the spherical Gaussian noise used in the construction of the weight process $\widetilde{W}\_t$. It is possible to replace the spherical Gaussian with a Gaussian noise having a non-diagonal covariance that reflects the geometry of the loss landscape. With this replacement, the dimension $d$ in the flatness term will be replaced by the trace of $\sum\limits\_{t=1}^T\kappa\_t$, where $\kappa\_t$ is the covariance matrix of the noise added at step $t$. Please refer to Neu'21[49] for a similar development.
>
> Regarding your question on comparing our bounds with stability-based SGD bounds (e.g., Hardt et al' 16[32]), we note that our bounds and those bounds are not directly comparable since they are derived from different conditions. Additionally, the stability-based SGD bounds (e.g. Hardt et al' 16[32]) often depends on the Lipschitz constant of the network. On one hand, the Lipschitz constant can hardly be determined accurately for a given neural network, making the bound difficult to compute. On the other hand, the Lipschitz constant captures only the "worst-case" gradients, indepedent of the data distribution. As a consequence, we expect such bounds to be more pessimistic than our bounds, which explicitly account for the data distribution.
>
>
> - Corollary 1: Assume we want to study the generalization error of a NN trained with SGD with respect to 0-1 loss. 0-1 loss is subG with parameter ¼. However, it is not continuous. Therefore, the bound in corollary 2 can’t be used for 0-1 loss function since it does not have second-derivative. Because the Gamma term is defined with respect to the loss function. How does this affect the conclusion of the paper on the impact of the flatness of the final solution on the generalization?
>
> **Response.**	We assume that the loss function considered in this paper is continuous (Line 89-90) and our generalization bounds are only valid for such losses. In case a discontinuous 0-1 loss is of interest, a common practice is to replace the loss with a continous surrogate loss function that is differentiable almost everywhere so that optimization with SGD is applicable. In this case, our bounds will be bounding generalization error defined using such surrogate losses. Nonetheless there may be room to further extend this work and bound the generalization error defined with respect to the 0-1 loss using the surrogate loss on which SGD is applied. The techniques developed in [E] may provide hints on how this can be done.
>
>
> - Diagonal covariance: Adding gaussian noise seems to be reasonable. However, why do not you consider the non-diagonal covariance?
>
> **Response.**	As mentioned above, it is indeed possible to use Gaussian noise with non-diagonal covariance matrix in the construction of the auxillary weight process. This should provide opportunities for developing even tighter bounds. But before we have a handle on how to construct those covariance matrices, such bounds will perhaps only
> serve a theoretical interest since they may not be computable numerically.
>
> - Stochastic Convex optimization: Assume that the loss function is convex, lipschitz, and smooth. How can you use this information to design the noise variances? It seems to me that this framework in this regime is much looser than Hardt et al results specifically because of the dependence on the dimension.
>
> **Response.**	Thank you for raising this question, which we find particularly interesting and inspiring. It is unfortunate that at present we do not have an answer. We speculate that the design of the noise covariance will require tracking a more general notion of gradient dispersion, likely a matrix, which  captures not only the "magnitude" of the dispersion as in the current definition of gradient dispersion, but also the "directions" of the dispersion.
>
>
>
>
> - Line 205: In your numerical results what do you mean by setting the sub-gaussian parameter to 0.1? If you use 0-1 loss then the subgaissian parameter is ¼.
>
> **Response.** As a usual choice for training DNN, we use cross entropy as the loss function in our experiment. To choose the variance proxy $R$, we first collected all the per-sample losses that were observed during training, then we used an Gaussian distribution to fit these loss values and we found that the standard deviation of the learned Gaussian  is between $0.1\sim0.3$. We thus select $R=0.1$. These details will be added in the Appendix.
>
> - It seems you use corollary 1 for empirical studies. As pointed out, this is not valid, it is not valid for 0-1 loss function.
>
> **Response.** The loss function used in the experiment is the cross entropy loss. Please refer to our earlier comments concerning 0-1 loss.
>
>
>
> [E] Cao, Yuan, and Quanquan Gu. "Generalization bounds of stochastic gradient descent for wide and deep neural networks." NeurIPS 2019.

---

> > ### Author Response · Authors · 2021-08-10
> > **To Reviewer dXbz (cont.)**
> >
> > - The conditional expectation notation in the paper should be revised. For instance in Eq (9) the notation is not correct and it makes it very difficult for readers to understand it. The outer expectation is over S,S’ but inside there is a conditioning on the specific realizations. The same error appears in the rest of the proof.
> >
> > **Response.** Thanks for carefully reading our proofs. We will revise our notations to improve their clarity and make them more accessible.
> >
> >
> >
> >
> > - I have difficulty understanding Eq(6) - Eq(10). Specifically, what do you mean by “we do not use an independent copy of $\tilde{W}$”? I believe the derivation in Neu’ 21 is correct (beginning of Page 8 in the V1 arxiv version of Neu’ 21). My understanding is as follows: P(W|S) is a kernel which shows the learning algorithm: specifically given a training set, it outputs a distribution on W. Therefore, when you have two independent training sets S and S’, they will generate two independent copies of W and W’ whose law are given by P(W|S) and P(W’|S’). Specifically Eq(8) in your paper is not correct. This is an extremely serious issue and I would like to hear your detailed response on this.
> >
> >  **Response.** Our proof is correct. The confusion might have arised from the sentence "we do not use an independent copy of $\widetilde{W}$", which is also correct in that context.
> >
> > We here elaborate on our proof in Eq. (6)- (10) with a slight change of notation for improved clarity.
> >
> > Consider two independent copies of training samples $S$ and $S'$, and let $W_t$ and $W'_t$ denote the weight processes of SGD on these two samples respectively. The context of this discussion may involve three auxiliary weight processes:
> >
> > $$\begin{align}
> > \widetilde{W}\_{t} &= \widetilde{W}\_{t-1}-\lambda G\_t+N\_t,\\\\
> > \widetilde{W}'\_{t} &= \widetilde{W}'\_{t-1}-\lambda G'\_t+N'\_t,\\\\
> > \widetilde{W}''\_{t} &= \widetilde{W}''\_{t-1}-\lambda G'\_t+N\_t,
> > \end{align}$$
> > where $\widetilde{W}\_{t}$ is generated from sample $S$ via $G\_t = g(W_t,B_t)$, $\widetilde{W}'\_{t}$ and $\widetilde{W}''\_{t}$ are both generated from another sample $S'$ via $G'\_t = g(W'\_t,B'\_t)$. But $\widetilde{W}'\_{t}$ and $\widetilde{W}''\_{t}$ do not share the same additive Gaussian noise. Specifically, the noise term $N\_t$ in $\widetilde{W}''\_{t}$ is the same noise variable in  $\widetilde{W}\_{t}$, whereas the noise term in $\widetilde{W}'\_{t}$ is another noise variable $N'\_t$ having the same distribution as $N\_t$ but independent of $N\_t$.
> >
> > First note that all three random variables $\widetilde{W}\_{t}$, $\widetilde{W}'\_{t}$ and $\widetilde{W}''\_{t}$ have the same marginal distribution, i.e., $P\_{\widetilde{W}\_t}=P\_{\widetilde{W}'\_t}=P\_{\widetilde{W}''\_t}$. Also note that $\widetilde{W}'\_t$ is an ``independent copy'' of $\widetilde{W}\_t$. But $\widetilde{W}''\_t$ is not independent of $\widetilde{W}\_t$  since they both depend on the same noise $N\_t$.
> >
> >
> > Neu' 21[49] uses $\widetilde{W}'\_t$ so that the independence condition of Lemma 4 in the V1 arxiv version of Neu’ 21[49] can be satisfied. In our proof, we use $\widetilde{W}''\_t$, dependent of $\widetilde{W}\_t$, hence the remark in question regarding not requiring an independent copy of $\widetilde{W}\_t$.
> >
> > The reason that our proof goes through is that our Lemma 4 does not require the independence condition insisted in Lemma 4 of Neu' 21 [49].
> >
> > To articulate our proof from Eq (6) to (10), recall
> > $$I(\widetilde{W}\_T;S) = \mathbb{E}\_{S}[\mathrm{D\_{KL}}(P\_{\widetilde{W}\_T|S}||P_{\widetilde{W}\_T})].$$
> >
> > Here replacing $P\_{\widetilde{W}\_T}$ by $P\_{\widetilde{W}'\_T}$ or $P\_{\widetilde{W}''\_T}$ does not change the value of the KL divergence, since these marginal distributions are the same. Note that the KL divergence is a functional depending only on the two distributions in its argument; as long as the two distributions are given, what random variables they correspond to is irrelevant to the evaluation of the KL divergence. Leveraging this fact, we now reproduce the proof from Eq (6) to (11) below (with revised notions), where we begin with replacing $P\_{\widetilde{W}\_T}$ with $P\_{\widetilde{W}''\_T}$ in the KL term.
> >
> >
> >
> > $$
> > \begin{align}
> > I(\widetilde{W}\_T;S)&=\int\_s\int\_w dP\_{\widetilde{W}\_T|S=s}(w)\log\frac{dP\_{\widetilde{W}\_T|S=s}(w)}{dP\_{\widetilde{W}''\_T}(w)}d\nu(s) \tag{1}\\\\
> >     &=\int\_s\int\_w dP\_{\widetilde{W}\_T|S=s}(w)\log\frac{dP\_{\widetilde{W}\_T|S=s}(w)}{\int\_s'dP\_{\widetilde{W}''\_T|S'=s'}(w)d\nu(s')}d\nu(s)\tag{2}\\\\
> >     &\leq \int\_{s,s'}\int\_w dP\_{\widetilde{W}\_T|S=s}(w)\log\frac{dP\_{\widetilde{W}\_T|S=s}(w)}{dP\_{\widetilde{W}''\_T|S'=s'}(w)}d\nu(s)d\nu(s')\tag{3}\\\\ &=\mathbb{E}\left[\mathrm{D\_{KL}}\left(P\_{\widetilde{W}\_T|S}||P\_{\widetilde{W}''\_T|S'}\right)\right] \tag{4}\\\\
> > &\leq\mathbb{E}\left[\mathrm{D\_{KL}}\left(P\_{\widetilde{W}\_{0:T}|S}||P\_{\widetilde{W}''\_{0:T}|S'}\right)\right] \tag{5}\\\\
> > &=\sum\_{t=1}^T\mathbb{E}[\mathrm{D\_{KL}}(P\_{\widetilde{W}\_{t}|\widetilde{W}\_{0:t-1},S}||P\_{\widetilde{W}''\_{t}|\widetilde{W}\_{0:t-1},S'})] \tag{6}.
> > \end{align}
> > $$
> > where $\widetilde{W}\_{0:t-1}$ denotes $\\{\widetilde{W}\_{0},\widetilde{W}\_{1}, \dots, \widetilde{W}\_{t-1}\\}$,  the expectation in (4) and (5) is taken over the joint distribution of $(S,S')$, and  the expection ${\mathbb E}$ in (6) is with respect to the joint distribution of $(S, S',\widetilde{W}\_{0:t-1})$.
> >
> >
> > In the above derivation, (1) is obtained by replacing  ${\rm D}\_{\rm KL}(P\_{\widetilde{W}\_T|S}||P\_{\widetilde{W}''\_T})$ with its definition; (2) is a rewriting of $P\_{\widetilde{W}''\_T}$ ; (3) follows from the Jensen's Inequality; (4) recovers the KL term from its defintion; (5) follows from the data processing inequality expressed in terms of KL divergence; and (6) is an application of the chain rule of relative entropy/KL divergence.
> >
> >
> > - Could you please elaborate on Eq (11)? Eq(10) follows from the monotonicity of the KL divergence or Data Processing. In Eq (11) you wanted to use the chain rule for KL. But the caveat here is that the surrogate algorithm \tilde{W} is NOT Markovian as shown in the figure on Page 4 of your paper. As a result, Eq (11) in its current form is NOT CORRECT. I would appreciate it if you could comment on it since it is an important step in the proof and it might affect the validity of the proof.
> >
> > **Response.** Thanks for asking this and we find a typo in our submitted version in Eq (11) and (12). Specifically, the correct expression in  Eq (11) should be $\sum\_{t=1}^T\mathbb{E}[\mathrm{D\_{KL}}(P\_{\widetilde{W}\_{t}|\widetilde{W}\_{0:t-1},S}||P\_{\widetilde{W}\_{t}|\widetilde{W}\_{0:t-1},S'})]$ where $\widetilde{W}\_{0:t-1}$ is $\\{\widetilde{W}\_{0},\widetilde{W}\_{1}, \dots, \widetilde{W}\_{t-1}\\}$. Note that this expression is the same as the Eq (6) in the above reproduction of the proof, using weight process $\widetilde{W}''\_t$.
> >
> >
> > This  result simply follows from the chain rule of the relative entropy (see Cover et al' 12[18, Theorem 2.5.3]).  Specifically (using the original notation),
> >
> > $$\begin{align}
> > &\mathbb{E}\left[\mathrm{D\_{KL}}\left(P\_{\widetilde{W}\_0,\widetilde{W}\_1,\dots,\widetilde{W}\_T|S}||P\_{\widetilde{W}\_0,\widetilde{W}\_1,\dots,\widetilde{W}\_T|S'}\right)\right]\\\\
> > =&\mathbb{E}\left[\mathrm{D\_{KL}}\left(P\_{\widetilde{W}\_0|S}||P\_{\widetilde{W}\_0|S'}\right)\right]+\mathbb{E}\left[\mathrm{D\_{KL}}\left(P\_{\widetilde{W}\_1|\widetilde{W}\_0,S}||P\_{\widetilde{W}\_1|\widetilde{W}\_0,S'}\right)\right]+\cdots+\mathbb{E}\left[\mathrm{D\_{KL}}\left(P\_{\widetilde{W}\_T|\widetilde{W}\_{0:T-1},S}||P\_{\widetilde{W}\_T|\widetilde{W}\_{0:T-1},S'}\right)\right]\\\\
> > =&\sum_{t=1}^T\mathbb{E}[\mathrm{D\_{KL}}(P\_{\widetilde{W}\_{t}|\widetilde{W}\_{0:t-1},S}||P\_{\widetilde{W}\_{t}|\widetilde{W}\_{0:t-1},S'})].
> > \end{align}$$
> >
> > Eq (12) had a similar typo. The following derivation should be used to replace Eq (12) and (13)
> >
> > $$\begin{align}
> > &\mathbb{E}\left[\mathrm{D\_{KL}}\left(P\_{\widetilde{W}\_t|\widetilde{W}\_{1:t-1},S}||P\_{\widetilde{W}\_t|\widetilde{W}\_{1:t-1},S'}\right)\right]\\\\
> > =&\mathbb{E}\left[\mathrm{D\_{KL}}\left(P\_{g(\widetilde{W}\_{t-1}-\Delta\_{t-1},B\_t)-\frac{N\_t}{\lambda\_t}|\widetilde{W}\_{1:t-1},S}||P\_{g(\widetilde{W}\_{t-1}-\Delta\_{t-1},B'\_t)-\frac{N\_t}{\lambda\_t}|\widetilde{W}\_{1:t-1},S'}\right)\right]\\\\
> >     \leq& \frac{\lambda^2\_t}{2\sigma\_t^2}\mathbb{E}\left[||g(\widetilde{W}\_{t-1}-\Delta\_{t-1},B\_t)-g(\widetilde{W}\_{t-1}-\Delta\_{t-1},B'\_t)||\_2^2 \right].
> > \end{align}$$
> > where the inequality above follows from Lemma 4, which requires no independence between $g(\widetilde{w}\_{t-1}-\Delta\_{t-1},B\_t)$ and $g(\widetilde{w}\_{t-1}-\Delta\_{t-1},B'\_t)$.
> >
> > The remaining part of the proof stays the same. -- Thank you for pointing to the error. Finally, we wish to add that even without going along this path of proof, Theorem 1 can be alternatively proved using Theorem 2 and Lemma 6.

---

### Official Review · Reviewer_ytrb · 2021-07-16

**Rating:** 7
**Confidence:** 4

**Summary:**

Improves existing information-theoretic generalization bounds for neural networks trained with SGD, and explores a new double descent phenomenon. Application consisting of GMP interesting but not competitive to best baselines.

**Limitations And Societal Impact:**

no social impact

**Main Review:**

This paper improves a bound in [49], removing a term therein by using a variant of the HWI. This bound is further improved from n^-1/2 to 1/n rate. These improvements, while not requiring a wholly new analysis, do seem to be significant improvements, and the authors discuss the interpretation and implications of the new bound thoroughly. I therefore recommend acceptance.

Line 61-62 can be removed, it is sufficient to mention additional material in the supplement in line.

**Time Spent Reviewing:**

NA

---

> ### Author Response · Authors · 2021-08-10
> **To Reviewer ytrb**
>
> Thank you very much for your positive comments. We will delete Line 61-62.

---

### Official Review · Reviewer_w2wb · 2021-07-19

**Rating:** 5
**Confidence:** 3

**Summary:**

This paper proves a new information-theoretic upper bound for the generalization of neural networks trained with SGD. The authors claim that the developed new bound provides new insights on the SGD and propose a new regularization scheme which can match the state-of-the-art result.


**Limitations And Societal Impact:**

This paper has no negative societal impact.

**Main Review:**

After reading this paper it seems that the main contribution is the improvement of the generalization bound compared to [49, Theorem 1.1]. In particular, the authors prove that the “local gradient sensitivity” term in [49] is unnecessary. However, it is not clear whether this improvement is significant. the authors may need to discuss which term in Lemma 3 could be the dominating one (i.e., comparing the magnitude of $\Psi(W)$ and $V(W)$). Besides, the authors should also include the bound in Lemma 3 in the experiment (e.g., Figure 1) to better interpret the improvements.

From Theorem 2 it can be seen that the generalization error becomes smaller when increasing the mini-batch size. Consider an extreme case where $b=n$, then the algorithm reduces to full-batch gradient descent. The authors did discuss a bit in Section 4, but a theoretical explanation could be better. Besides, why not directly compare the proved generalization bound or test accuracy for different batch sizes as that could be more related to the theory of this paper.


**Time Spent Reviewing:**

4 hrs

---

> ### Author Response · Authors · 2021-08-10
> **To Reviewer w2wb**
>
> Thank you for your contructive comments. Our responses follow.
>
>
> - However, it is not clear whether this improvement is significant. the authors may need to discuss which term in Lemma 3 could be the dominating one (i.e., comparing the magnitude of $\Phi(W)$  and $\mathbb{V}(W)$). Besides, the authors should also include the bound in Lemma 3 in the experiment (e.g., Figure 1) to better interpret the improvements.
>
> **Response.** As a short answer, removing the gradient local sensitivity term $\Phi(W)$ significantly improves the bound. As we will give more details below, we would like to first remark that in a practical training scenario, the trajectory term in the bound can be reduced by more than three orders of magnitude (i.e., reduced to 0.1 percent) when the local sensitive term is removed.
>
>
> First note that computing $\Phi(W)$ requires significant computation complexity. This is because computing $\mathbb{E}_{\zeta}g(w+\zeta,z)$ will need additional forward and backward passes during training and involve the complexity of drawing samples for $\zeta$. Moreover, the estimation of $\Phi(W)$ requires explicitly defining the noise variance $\sigma^2$. This contrasts our bound, for which the the optimal $\sigma^2$ has a closed-form expression (see line 203-204). This has been the reason that we did not plot the bound in Lemma 3 when submitting the paper.
>
> Upon seeing your comments, we agree that it is important to quantify the improvement of our bounds numerically and hence made the following experimental efforts comparing $\Phi(W)$ and $\mathbb{V}(W)$ on MNIST.
>
> Note that the optimial $\sigma$ for our bound is around 0.0001 for MLP with 512 hidden units, we thus take $\sigma=0.0001$ in this comparison. To compute $\Phi(W)$, we apply Monte-Carlo estimation where $20$ noise samples are drawn for each iteration. We train MLP for 30 epochs and observe that $\sum_{t=1}^T\Phi(W_{t-1})$ is over 100 times of $\sum_{t=1}^T \mathbb{V}(W_{t-1})$. Factoring in the impact of batch size (60), the trajectory term in the bound of Lemma 3 is about 6000 times of that in our bounds.
>
> This remarkable improvement should come at no surprise, since $\Phi(W)$ monotonically increases with training epochs (noting that $\zeta$ has the cumulative variance $\sum_{i=1}^{t-1}\sigma_i^2I_d$) while $\mathbb{V}(W)$ appears monotonically decreasing (see also Figure 3). Also note that when the loss function is $\mu$-smooth, it is possible to show theoretically that the worst $\Phi(W)$ can be as large as $2\mu^2\sigma^2Td$.
>
> Finally on a separate note, we wish to remark that removing the gradient local sensitivity term $\Phi(W)$ not only makes the bound tighter, it also makes its evaluation much simpler.
>
>
>
>
>
>
>
> - From Theorem 2 it can be seen that the generalization error becomes smaller when increasing the mini-batch size. Consider an extreme case where $b=n$, then the algorithm reduces to full-batch gradient descent. The authors did discuss a bit in Section 4, but a theoretical explanation could be better. Besides, why not directly compare the proved generalization bound or test accuracy for different batch sizes as that could be more related to the theory of this paper.
>
> **Response.** We first like to note that Theorem 2 does not imply that increasing batch size will reduce the generalization error. This is due to the flatness term in the bound, which tends to increase with increasing batch size, as shown in Figure 2 and other previous empirical observations [33,37,76]. Our current information-theoretic bounds do not explian why small batch sizes will return flat minima. Some recent works based on backward error analysis (e.g., [C,D]) may however provide analytic tools to investigate this question. In particular, similar terms resembling gradient dispersion and the flatness also appeared as the regularizers in [C, D]. It is our interest to explore this direction, but this appears a separate topic in its own right.
>
> In the revision, we will make an effort including an experimental comparison of our bound and the test accuracy across different batch sizes.
>
> [C] Barrett, David, and Benoit Dherin. "Implicit Gradient Regularization." ICLR 2021.
>
> [D] Smith, Samuel L., et al. "On the Origin of Implicit Regularization in Stochastic Gradient Descent." ICLR 2021.

---

### Official Review · Reviewer_Bv3T · 2021-08-02

**Rating:** 5
**Confidence:** 3

**Summary:**

This paper studies the generalization of neural networks trained using stochastic gradient descent (SGD) using information-theoretic bounds. It builds upon existing work that bounds the generalization error using the average square root of the mutual information of the weights with each training datum and related work by Neu 2021 [49] that refines this bound using quantities that correspond to the local gradient sensitivity, the variance/dispersion of the mini-batch gradients, and a stability term that captures the loss at the terminal weights on independent test data. The main contributions of this manuscript are as follows.

1. The paper introduces a coupled auxiliary weight process which refers to the weights updated using additive noise in the gradient descent direction calculated by SGD using the original weights. This allows the authors to study the mutual information of the descent direction with a particular datum in Lemma 5.
2. Theorem 1 establishes that the gradient sensitivity term can be eliminated using an application of Otto and Villani’s HWI inequality that relates the entropy of the weight posterior with the Wassserstein distance and the Fisher information. Theorem 2 and Corollary 1 expand on this result. The main conclusion is that the generalization error can be broken down into two terms, one that depends upon the ratio of the learning rate and batch-size and another that depends upon the stability term referred to above (bounded in Corollary 1 using the trace of the Hessian).
3. The bound in Theorem 2 is verified using a fully-connected network on MNIST and a convolutional network on CIFAR-10.
4. The paper also develops a loss function that training neural networks that averages the loss of the network across different perturbations of the weights by Gaussian noise. This new loss function is evaluated on MNIST, CIFAR-10 and CIFAR-100.

**Limitations And Societal Impact:**

Adequate.

**Main Review:**

Main comments

1. This paper is very closely related to [49]. It is understood that this reference is, as yet, unpublished. The authors should however discuss the novelty compared to respect to this existing work.
2. Line 214: Arguably, the red line in Fig. 1 a,b does not decrease but rather settles down to a steady value as the model size increases. Is this an experimental artifact?
3. How do you explain the fact that the upper bound in Theorem 2/Lemma 6 increases with T, i.e., is larger for longer trajectories? In practice, the generation gap clearly remains bounded as the number of training epochs increases. The authors come close to discussing this around Line 230 but it is necessary to expand upon this for bounds such as those in this paper.
4. It is possible that the complex relationship between the learning rate, number of training steps T and the upper bound stems from the fact that the gradient dispersion term is not “normalized“. This term (Line 123) does not distinguish between the gradient dispersion being small somewhere at the beginning of training (say all per-sample gradients have similar magnitudes and point in similar directions) versus the term being small at the end (which would indicate training finding a critical point that is close to the critical points of the per-sample loss functions). It may be possible to separate out these two phenomena.
5. The above point also calls into question the importance of the “double descent of gradient dispersion” phenomenon explored in Figure 3. Indeed, for noisy samples, it has been seen that there is a long phase in training where the per-sample gradients show a strong mismatch and the training/test loss plateaus (https://arxiv.org/abs/2101.06509). It is unclear what the implication of this double descent observation is and how it is related to the rest of the paper.
6. I have concerns about the experiments in Table 1. First, such loss functions have been used numerous times in prior work (the authors do cite papers on flat minima so they are aware of algorithms like https://arxiv.org/abs/1611.01838, https://arxiv.org/abs/1609.04836 etc.); so this loss is not really novel. Second, networks trained with this new loss generalize about as well as algorithms like AMP, so this experiment is more of a sanity check than a new contribution. It would be good for the manuscript to demarcate the new contributions explicitly in the introduction.

In conclusion, there are interesting theoretical parts in this manuscript but (i) it is difficult to ascertain whether the theory is novel enough to merit publication, and (ii) the experiments seem to be essentially disjoint of the theory. Perhaps, the results in Theorem 3 could be fleshed out and added to the main text.

**Time Spent Reviewing:**

4

---

> ### Author Response · Authors · 2021-08-10
> **To Reviewer Bv3T**
>
> We thank you sincerely for providing valuable input to this work. Below we discuss your comments and concerns in detail.
>
>
> 1. We first like to note that [49] has been recently accepted by COLT 2021.	 The novelty of this work, relative to [49], was discussed in Introduction. Specifically, our bounds remove an unnecessary term in  [49] and is further tightened by using the individual mutual information bound in place of the sample-wise mutual information bound in [49]. In addition, removing the unnecessary term makes the bound easier to verify numerically. Some practical implications of our bounds, for example, dynamic gradient clipping and GMP, are also discussed.
>
>
> 2. Overall the red curve (our bound in Corollary 1) decreases with the model size, following the same trend as the blue curve (the estimated true bound). The fact that the decreasing trend of the red curve appears to settle is due to that the blue curve flattens to near zero and that the difference between the two curves does not vanish. The latter fact in part results from a difference between the bound in Corollary 1 (the plotted bound) and the bound in Theorem 2. In Theorem 2, the flatness term is $\mathbb{E}{[\gamma(W_T,S)-\gamma(W_T,S')]}$ whereas in Corollary 1 the flatness term is  upper-bounded by $\mathbb{E}{[\gamma(W_T,S)]}$ (which further is expressed as the trace). For this reason, the blue curve is expected to stay bounded above zero. Finally, we choose to plot the bound in Corollary 1 rather than that in Theorem 2 since we wish to use only the training data to in computing the bound.
>
>
> 3.	We note that the bound in Theorem 2 needs not to always increase with $T$. This is because as training continues, the gradient dispersion may eventually become zero (as shown in Figure 3), thus they do not contribute to the sum in the trajectory term of the bound. But before training reaches zero gradient dispersion, the trajectory term does depend on the number of training iterations and one expects that rapidly forcing training to converge helps to reduce the generalization error, a phenomenon observed in a previous work ``Train faster, generalize better'' [32].
>
>
> 4 & 5. Thank you for commenting on the ``unnormalized'' nature of our notion of gradient dispersion and for pointing to reference [A]. Indeed, our definition of gradient dispersion makes no attempt to separate the contributions of gradient norms and their alignment, which are shown in [A] to jointly affect the SGD training dynamics. While it is certainly useful to separate the influence of gradients from these two aspects for understanding the training dynamics of SGD, the results of our work appear to suggest that as far as generalization is concerned, the compressed measure in terms of gradient dispersion and how it gets accumulated during training might be nearly (if not completely) adequate.  We now elaborate on this point.
>
> First theoretically as seen in Lemma 5, controlling the gradient dispersion can control the conditional mutual information between the gradient signal and the current batch of examples. When this mutual information is made smaller, the updates by SGD depend less on the specific training sample, giving rise to an algorithmic stability and allowing the model to generalize better.
>
> Second, as shown in Figure 3, gradient dispersion appears to well correlate with the gap between the training curve and the testing curve (see, Line 241 - 250).
>
> After reading your comments, we performed experiments investigating the impact of gradient norm and gradient alignment on the generalization error (the gap between training accuracy and testing accuracy). Specifically, in each training batch, we measure the average of the gradient norms over the samples in the batch. We also measure the cosine similarity between every pair of gradients in the batch and compute their average, which we use as a measure of gradient alignment. For MLP trained on MNIST, we observe that gradient norm follows a similar trend as gradient dispersion, i.e.,  decreases with training epochs in the clean dataset and exhibits a double-descent phenomenon in the noisy datasets. Therefore, like  gradient dispersion, gradient norm also appears to mark the three phases of training as we discuss in the paper.
>
> On the other hand, the average cosine similarity reveals a different dynamics from gradient dispersion. In particular, on noisy datasets, the cosine similarity first increases and then decreases, where the turning point appears before the transition from phase 2 to phase 3 and no clear indication of the transition from phase 1 (generalization) to phase 2 (memorization) is observed.
>
> At this end, we conclude that the trend of gradient dispersion and its corresponding influence on generalization are dominated by the gradient norm. Although gradient alignment reveals some insight into the training dynamics as discussed in [A], it appears less relevant to the generalization error.
>
>
> >- It is unclear what the implication of this double descent observation is and how it is related to the rest of the paper.
>
> As discussed earlier, the theoretical results of this paper suggest that gradient dispersion plays an important role in generalization, since it contributes to the trajectory term in our bounds (see also Lemma 5). This motivates us to investigate the behaviour of gradient dispersion experimentally.
>
> The "double descent" phenomenon of gradient dispersion observed in our experiments shows that the behaviour of gradient dispersion is closely related to the generalization behaviour of the network at different training stages. This to an extent validates our theoretical derivation. On the other hand, in phase 2 (memorization phase) of the SGD training, a large gradient dispersion appears to merely correlate with memorization of noisy data. One implication of this is that one may design regularization schemes by controlling the gradient dispersion, for example, using an approach similar to "dynamic gradient clipping" studied in this paper or exploiting early stopping in a way similar to that studied in [B] -- note that in [B] a metric referred to as "gradient disparity" was studied, which shares great similarity with gradient dispersion of this paper.
>
>
>
>
>
>
>
>
> 6. We agree that GMP may well serve as a sanity-check. But to the best of our knowledge, we are not aware of any previous work that explicitly uses the same loss function. Most literature exploiting noise for regularization add random perturbations to input or feature maps, but in GMP, the perturbation is added to the model parameters, serving as an "average pooling" in the parameter space. The closest approach to GMP in our knowledge is Entropy-SGD [16] (which you also point to), where a different loss function sharing a similar "flatness-promoting" property is constructed. If you are aware of any other work that uses a loss function similar to the GMP loss function, we would greatly appreciate a pointer (note that the other reference you point to does not explicitly construct a new loss function).  In that case, we will be happy to tone down our contribution in introducing GMP, and solely regard it as a sanity check.
>
>
>
> 7. >- Perhaps, the results in Theorem 3 could be fleshed out and added to the main text.
>
> Theorem 3 indeed points to directions to further this work. However, as we are mostly concerned with developing computable bounds for the generalization error, we are yet to find a computatble characterization (or tight bounds) of the contraction coefficient $\eta_{i, \tau}$. That appears to involve additional machinery and require significant effort. Without such a characterization, we feel including Theorem 3 in the main text may present unnecessary complexity for the reader.
>
>
> [A] Feng, Yu, and Yuhai Tu. "Phases of learning dynamics in artificial neural networks: with or without mislabeled data." arXiv preprint arXiv:2101.06509 (2021).
>
> [B] Forouzesh, Mahsa, and Patrick Thiran. "Disparity Between Batches as a Signal for Early Stopping." arXiv preprint arXiv:2107.06665 (2021).

---

### Decision · Program_Chairs · 2021-09-27

**Decision:**

Reject

**Comment:**

This paper proves an information-theoretic generalization error bound of neural networks trained by SGD. The main concern of the reviewers is about the novelty and significance of the results derived in this paper because this paper is heavily built upon recent work in COLT 2021. Even after the author response, this paper does not receive sufficient support from the reviewers. Therefore, I recommend rejection.